# Construction and Experimental Analysis of a Multipurpose Robotic Fin Ray Gripper for Manipulator Robots

**DOI:** 10.3390/s25185782

**Published:** 2025-09-17

**Authors:** Anselmo Rafael Cukla, Rafael Crespo Izquierdo, Lucas Strapazzon, Joaquín Ezequiel Taverna, Claudenir Rocha Alves Filho, Sergio Omar Lapczuk, Jorge Antonio Szydlowski, Solon Bevilacqua, Daniel Fernando Tello Gamarra

**Affiliations:** 1Centro de Tecnologia, Departamento de Processamento de Energia Elétrica, Universidade Federal de Santa Maria, Santa Maria 97105-900, RS, Brazil; anselmo.cukla@ufsm.br (A.R.C.); daniel.gamarra@ufsm.br (D.F.T.G.); 2Escola de Engenharia, Departamento de Engenharia Mecânica, Universidade Federal do Rio Grande do Sul, Porto Alegre 90010-150, RS, Brazil; crespo@ufrgs.br; 3Centro de Tecnologia, Programa de Pós-Graduação em Engenharia Mecânica, Universidade Federal de Santa Maria, Santa Maria 97105-900, RS, Brazil; lucasstrapazzonpf@gmail.com; 4Facultad de Ingeniería, Departamento de Ingeniería Mecatrónica, Universidad Nacional de Misiones, Obera CP 3360, Misiones, Argentina; joaquin.taverna@fio.unam.edu.ar (J.E.T.); sergio.lapczuk@fio.unam.edu.ar (S.O.L.); jorge.szydlowski@fio.unam.edu.ar (J.A.S.); 5Centro de Tecnologia, Curso de Engenharia da Computação, Universidade Federal de Santa Maria, Santa Maria 97105-900, RS, Brazil; 6Departamento de Engenharia de Produção, Faculdade de Ciências e Tecnologia, Universidade Federal de Goiânia, Goiânia 74690-900, GO, Brazil; solon.bevilacqua@ufg.br

**Keywords:** fin ray, flexible end-effector, gripper, robotic manipulation, finite element method

## Abstract

This article presents a methodology for estimating the gripping forces in a Fin Ray-type gripper, based on the integration of experimental and computational approaches. The development and validation methods includes (1) mechanical modeling and material selection; (2) experimental tests to relate FG finger displacement to maximum applied force using a load cell; (3) validation of the computational model through finite element method (FEM) simulations in ABAQUS using experimental data; and (4) experimental analysis of the FG handling a chicken egg, with the FEM determining the stress applied to the egg. The computational results showed a maximum stress of approximately 7 MPa on the egg, with no signs of damage, demonstrating the FG’s suitability for handling delicate objects in both the experimental and computational procedures, thus enabling safe object handling without causing damage. This work advances research on Fin Ray-type flexible end-effectors, emphasizing their utility in manipulating fragile objects without requiring complex force and pressure control algorithms.

## 1. Introduction

End-effectors, or robotic grippers, are devices that define the functionality of manipulator robots in handling tasks, representing a topic of great relevance to industry and other sectors of society. End-effectors allow robots to perform tasks considered complex or risky for humans in hazardous environments, such as handling heavy or radioactive objects. Furthermore, end-effectors can be designed for specific tasks, such as welding, handling parts, and more, which are particularly useful in automotive, aerospace, and other industries.

According to [1,2,3,4,5,6], end-effectors can be classified into four types: rigid grippers, tools, non-mechanical grippers, and mechanical grippers. Rigid grippers have no relative movement in their terminal elements and are commonly used for loading and unloading pallets, handling boxes, etc. Tools are employed in tasks such as welding and painting. Non-mechanical grippers use devices such as magnets and suction systems to enable object handling. Mechanical grippers have a gripping mechanism, also known as fingers, that allows direct contact with objects. This type of gripper can be classified as rigid or flexible, depending on the mechanical properties of its elements.

Rigid grippers are widely used in robotic applications involving object manipulation and are often found in industrial and educational environments. One of the main advantages of this type of gripper is its ability to handle small objects in repetitive tasks on assembly lines, such as nuts, bearings, screws, electronic components, etc. [7]. However, this type of gripper requires efficient algorithms for control, sensing, and instrumentation, which may hinder quick and practical implementation depending on the application. Recent studies on the development and application of rigid grippers show their efficiency in manipulating fragile objects with precise movements, analyzing the use of pressure and force sensors, material science, manufacturing techniques, and control algorithms in various applications involving conventional end-effectors [8,9,10].

Furthermore, flexible grippers offer several advantages in handling fragile objects, as they eliminate the need for highly reliable algorithms or precise position controls. This is particularly useful in manipulating everyday items such as fruits, food, or glass containers, ensuring efficiency and safety during object handling without requiring complex systems. The main research areas on this type of gripper focus on finger modeling, material selection, force analysis on the object using finite element methods, and more, investigating potential improvements in force control for handling various types of objects. Flexible grippers include Fin Ray-type grippers, which are a specific example of flexible designs. While both rigid and flexible grippers are widely used in specific object manipulation applications, this study aims to contribute to research investigating potential improvements in force control for handling both rigid and fragile objects in domestic and industrial tasks [8,9,11].

Fin Ray-type grippers offer passive adaptability and can conform to objects of various shapes [12,13,14]. This adaptability makes them suitable for grasping delicate objects without the need for complex control algorithms or sophisticated sensor instrumentation. However, Fin Ray grippers are not ideal for handling very small objects, unlike rigid grippers. Additionally, it has been demonstrated that flexible grippers with Fin Ray-type structures can also be efficiently used in robot locomotion on uneven surfaces, highlighting their versatility in robotic applications [11]. Although rigid grippers may provide greater gripping force, the advantages of flexibility, passive adaptability, and the potential integration of sensors make flexible Fin Ray grippers a promising alternative, particularly for handling fragile objects or tasks requiring conformity to the object [7,11,12,13,14].

### 1.1. Related Works

Recent studies on flexible robotic grippers (Fin Ray) have made significant advances in robotics, as these devices facilitate the manipulation of small, fragile, and irregularly shaped objects. Flexible end-effectors enable manipulator robots to perform precise movements in their tasks without exerting excessive force on objects, demonstrating high adaptability to different kinematic configurations. Such characteristics are essential in industries like pharmaceuticals and food processing [15,16,17].

The mechanical design of an end-effector involves the kinematic analysis and synthesis of the gripper mechanism, the dynamic analysis of forces, design studies, and prototype fabrication. These design steps should consider maximizing the contact area between the gripper elements and the object and the potential incorporation of sensors and actuators to adapt to objects of different shapes and sizes. In [3], a chronology of flexible gripper evolution is presented, categorizing them into three technology types: (A) actuation, (B) controlled stiffness, and (C) controlled adhesion. The author analyzes the material technologies used in different types of flexible gripper mechanical structures. The study highlights that the Fin Ray-type gripper, categorized under group C for stiffness or controlled deformation, is one of the most commonly employed for manipulating a wide variety of items such as spheres, eggs, vegetables, flower bulbs, toys, and more. These characteristics have motivated recent studies on this type of gripper for domestic use, involving the manipulation of objects with diverse shapes, sizes, and weights.

There are several recent studies proposing the development of Fin Ray-type flexible grippers. The main approaches proposed by the authors focus on the design of grippers for handling objects with various geometries (cylinders, spheres, cubes, and irregular shapes), studies to establish the largest contact area between the gripper and the objects avoiding possible deformations or breakage and simulations and experiments to propose the use of flexible grippers in the food industry or domestic robots.

In [18], a Fin Ray-type gripper was proposed for handling fruits, vegetables, and fragile products. The author designed the separation of the internal ribs with an increasing angle effect, trying to maintain the same spacing between the gripper’s ribs. The results obtained in this research showed that the orientation of the gripper’s ribs significantly increased the objects’ support force. A structural analysis of the stresses on the gripper’s joints and elements was not performed. In [19], an evolution of the initial proposal was presented. The author designed an actuator with variable stiffness, using the layers and the orientation angle of the ribs in the structure. Experimental tests and numerical simulations demonstrated that the variable stiffness capability not only increases effectiveness in handling different types of objects but also reduces the risk of damage to fragile objects.

In [20], a passive Fin Ray-type gripper was developed, in which the opening mechanism is driven by a lead screw and a motor. The author conducted experiments showing excellent adhesion of the proposed gripper in handling the test specimen. Additionally, the author experimentally evaluated the maximum weight supported by the gripper in critical positions, but no structural analysis of the gripper elements was performed.

In [21], a three-finger Fin Ray-type gripper was developed, printed using 3D technology and flexible material. An experimental procedure was conducted to measure the force exerted by the gripper on the object, but no structural analysis of the stresses on the joints and gripper elements was carried out. Similar studies addressing the development of flexible Fin Ray-type grippers for manipulation tasks include [14,16,22,23,24,25].

The studies mentioned here demonstrate that Fin Ray-type grippers are suitable devices for handling objects of various shapes and sizes. There are numerous approaches in the literature exploring Fin Ray-type grippers. Many studies aim to develop a passive mechanical actuator that is rigid and easy to construct for delicate object manipulation tasks. Additionally, some studies apply specific methods, such as finite element analysis, statistical modeling, and mathematical deformation analysis, among others, to establish or predict gripper deformation and provide dynamic information for the control system.

Fin Ray-type effectors are ideal for handling delicate objects. To contribute to studies investigating the application of mechanical grippers in handling fragile objects, this work aims to ensure the safe handling of chicken eggs and validate the forces applied to them. This study aims to compare the results obtained experimentally and computationally with the mechanical properties of eggs described in the literature. Although the Fin Ray effect and internal structural variations have been extensively studied in the literature, as in [26,27], this work contributes to the development of a method for estimating the force applied to an object. The proposed approach was analyzed and validated through an experimental procedure using a chicken egg as the study object. The intention is not to surpass advanced solutions, such as [28], but rather to offer a validation study for the safe manipulation of fragile objects. Nevertheless, minor improvements to the gripper are being proposed to enable more appropriate grasping movements.

Previous works, such as [29], show that physical properties of eggs, such as length, surface area, and breaking force, vary according to weight (medium: 52.23 g; large: 57.38 g; extra-large: 64.07 g; jumbo: 71.58 g). The longitudinal stress on the z-axis for rupture decreases with increasing weight, while specific deformation and rupture energy increase. In [30], quasi-static tests indicated that the maximum force before failure was 49 N, with the numerical model validated by finite element analysis in ABAQUS software (version 2024). Meanwhile, ref. [31] reported forces up to 750 N under distributed load, influenced by geometry and microstructure. This work uses these data to perform simulations and experiments, qualitatively validating the suitability of the proposed gripper for safe egg handling.

Although the finite element method (FEM) is widely employed in the design of flexible robotic grippers, relatively few studies incorporate experimental data as explicit boundary conditions to rigorously validate computational models and assess the safety of handling fragile objects. In this study, a methodological framework is proposed that integrates direct grip-force measurements with high-fidelity FEM simulations to calibrate and validate the computational model, thereby enabling more reliable estimation of the forces exerted during the grasping process of Fin Ray-type flexible robotic grippers.

In contrast to conventional approaches, the proposed method facilitates quantitative safety assessment in the manipulation of delicate objects by comparing simulated stress distributions against experimentally determined mechanical properties, as demonstrated in the safe handling of eggs. Recent works, including [32,33], underscore the relevance of combining FEM analyses with experimental measurements to enhance the predictive accuracy of force and deformation in Fin Ray and other soft grippers, further evidencing the significance of this integration for advancing the state of the art.

### 1.2. Objectives and Paper Organization

This research aims to demonstrate the steps for developing and validating a flexible robotic gripper (FG) of the Fin Ray type, based on the concepts presented in [7,14,20,34,35]. The objectives of this study include (a) the development and mechanical construction of the FG; (b) the validation of the FG’s maximum forces through experimental tests and simulations using the finite element method; and (c) the demonstration of its effectiveness in handling delicate objects through experimental and computational tests, using chicken eggs as a case study.

A central contribution of this work lies in the development of an integrated framework that combines FEM simulations with experimental data to provide robust and qualified estimations of the gripping forces exerted by a Fin Ray-type flexible gripper. Unlike traditional approaches that rely solely on isolated FEM modeling or require sophisticated instrumentation for force control, our method offers a practical and transferable alternative that facilitates the assessment of safety in the handling of fragile objects. This approach enables application across diverse contexts, offering a balance between simplicity and technical rigor.

The paper is structured as follows: Section 2 describes the methods for developing the proposed Fin Ray-type FG and presents the results of finite element simulations and experimental tests to evaluate the stresses on the gripper when handling objects. Section 3 presents the simulation results obtained by applying the proposed methods. Section 4 provides a detailed discussion and interpretation of these results. Finally, Section 5 presents the conclusions of this study.

## 2. Materials and Methods

The Fin Ray-type FG was developed with the objective of minimizing the risk of damage to fragile and rigid objects, enabling the execution of complex manipulation tasks in domestic, industrial, and other environments. The proposed methods include the construction and selection of FG components, as well as a feasibility study for its future application in handling fragile objects. This study involves determining the forces exerted by the FG on the object through a qualitative comparative analysis between simulations using the finite element method in ABAQUS and experimental procedures, which will be presented below. The proposed steps for the FG’s design and validation are shown in Figure 1.

### 2.1. FG Development

The constructive characteristics of the components and the functionalities of the FG were developed using SolidWorks software (version 2023). Below, the criteria adopted for defining the FG components and the force analysis applied by the FG on the object are presented. It is important to highlight that the methodology presented below for defining the gripper geometry was established based on various CAD simulation tests involving the manipulation of different objects (e.g., pens, apples, cups, etc.). From these simulations, the current structure was determined to be the most suitable. For both the simulation and experimental tests, a chicken egg was used, as it is a commonly manipulated object by domestic robots and has well-documented structural and fracture data [29,30,31].

#### FG Components

The FG design was conceived with mechanisms composed of rigid and flexible materials, incorporating a transmission system for opening and closing movements, along with a stepper motor. The FG features two fingers capable of adapting to the object, providing better grip. Figure 2 illustrates the main components of the proposed FG.

In Figure 2, the fingers are fixed to quick-release rails, which were designed to allow for the replacement of the finger assembly, if necessary, according to the characteristics of the object to be handled. The main mechanism of the FG was designed with rigid material, which supports the fingers and the mechanisms responsible for the opening and closing movement of the FG. A lead screw was added to this mechanism, responsible for converting the rotational movement of the motor into the linear movement of the FG.

For the construction of the flexible fingers, internal ribs were used. This type of solution allows for maximizing and minimizing deformation at the center and the tip of the fingers, respectively. Figure 2 shows the ribs of the fingers fixed to the quick-release rails. It is important to highlight that this deformation capability increases the contact area between the fingers and the object, resulting in better grip between them.

The triangular shape at the tip of the FG aims primarily to prevent deformation at the edges of the fingers, facilitating the handling of small objects such as eggs, pens, nuts, and others. Furthermore, the fingers have vertical centralizers, specially designed to hold cylindrical objects, such as bananas, forks, and spoons, in a longitudinal position. This not only protects the objects from excessive pressure but also reduces the risk of damage, maximizes the contact area, and decreases the pressure exerted on the object. A discussion on the force distribution between the FG and the object is presented in detail in Section 2.2 and Section 2.3.

The construction of the fingers and the FG drive mechanism was carried out using additive manufacturing techniques. The components of the drive mechanism, such as the FG base, motor support, rigid fingers, and finger mounting arms, were manufactured using Modified Glycol-Polyethylene Terephthalate (PETG). According to [36], the tensile test of this material indicates an approximate force of 160 kg and a deformation of 7.2 mm in a test specimen manufactured according to ASTM D 638 guidelines [37]. The choice of this material is due to its impact resistance and lower tendency to break [38,39], characteristics necessary for the FG mounting assembly.

The fingers were printed using Thermoplastic Polyurethane (TPU) filament, as the characteristics of this material, in terms of friction, deformation, and strength, are suitable for gripping objects with different geometries. According to [40], this material can withstand up to 53.7 MPa of tensile stress and a rupture elongation of 318%, according to the DIN 53504 standard [41].

The transmission system consists of an M8 screw (lead screw) and an M8 nut. To integrate these components with the rest of the assembly, an aluminum part is used to ensure rigidity and maintain the proper position. The motor chosen for the FG drive system was the MX-106 model from Dynamixel (ROBOTIS, Seoul, South Korea), which allows for the control of position and speed parameters. The main components of the proposed actuator are presented in detail in Figure 3.

Although several recent studies have explored advanced aspects of the design, optimization, and control of Fin Ray grippers, this work distinguishes itself by proposing a simplified and practical framework that seeks to balance functionality with ease of implementation. In contrast to approaches that emphasize sophisticated structural optimization methods [26,27] or complex integrations of sensors and active control [28], our focus lies in the integrated experimental and computational development aimed at ensuring the safe manipulation of fragile objects through an accessible and rapidly deployable methodology. This combination of robust experimental validation with FEM simulations specifically oriented toward preserving the integrity of the handled object aims to broaden access to cost-effective solutions, particularly for domestic and simple industrial environments.

To implement this approach in practice, the internal rib architecture of the FG was developed through an iterative process based on CAD modeling, supported by simulated experiments involving a variety of objects (e.g., pens, fruits, and containers). Although no formal parametric optimization study was conducted, the adopted structural modifications—including the symmetric distribution of ribs, the incorporation of grooves in the central regions, and the half-moon profile of the central rib (Figure 2)—were qualitatively validated for their ability to enhance conformity during grasping and to promote a more homogeneous distribution of the applied forces.

This design strategy aimed to ensure a functional and safe prototype for the manipulation of fragile objects while maintaining simplicity in construction. A detailed quantitative analysis of these improvements constitutes an important opportunity for future research.

### 2.2. Analysis of the Forces in the FG

The forces in the FG were determined through experimental procedures. The tests consisted of measuring the force applied by the actuator onto a load cell system. This analysis aims to determine the force between the FG and the object for later validation with the FEM. Figure 4 shows the main selected components of the developed test bench.

The main components used in this bench are as follows:SAE 1020 steel support for fixing the load cells;Two 20 kg nominal load cells, made of aluminum alloy, with an operating voltage of 5 V;Two half-spheres printed using 3D technology, made of PETG, serving as contact surfaces between the FG and the load cell;An Arduino Uno microcontroller (Arduino, Scarmagno, Italy) with 1 ms reading time and a communication baud rate of 57,600;A HX711 module for conditioning the load cell signals, with an analog-to-digital converter for scales;A 248 g standard mass for load cell calibration.

It is important to note that the tests were conducted using only one active load cell for the measurements, as there is no difference in the force applied by both fingers of the FG due to its construction characteristics. Thus, the second load cell was used to keep the other finger of the FG stable and under the same load conditions. The analyses presented in this section will be evaluated alongside the computational simulation studies by the FEM, discussed in Section 3, to validate the forces generated by the FG on the object. Although strain gauges were not implemented in this study, their use is being considered for future work in order to provide real-time force feedback.

### 2.3. Analysis by FEM

The FEM aims to investigate the relationship between the forces generated and the displacements caused when the FG comes into contact with the object. This analysis was conducted to validate the experimental analysis of the forces (see Section 2.1), thus establishing a method to estimate the force exerted by the FG on any object. The entire FEM implementation was carried out in the ABAQUS software. The verification and validation of the simulation tests were based on the ASMEV&V 10-2006 standards [42] and considerations on overload and structural collapse [43], although this work does not perform destructive analysis, and all tests are within the linear deformation range of the FG’s structural materials.

Regarding the FG model, it was decided to carry out the simulations using a 1/4 section of it, facilitating quick results and then applying boundary conditions to obtain the complete model. For meshing this model, considering hyperelastic materials, tetrahedral elements with an average size of 1 mm were used, both for the FG and the objects under analysis.

Regarding the movement restrictions of the FG fingers, a displacement path of 26.76 mm was defined for each finger during the closing motion in the horizontal direction toward the object, as illustrated in Figure 5. This displacement value corresponds to the average displacement observed during the experimental squeezing test of the half-spheres shown in Figure 4. For this simulation, a linear motion was considered, and the movement is performed solely by the base of the FG finger.

As for the contact points, they occur tangentially between the object surface and the finger, with a friction coefficient of 0.2 considered. The CAD model of the FG and the restrictions used in the proposed FEM analysis are illustrated in Table 1.

The maximum displacement applied by the FG was defined as 26.76 mm, producing a force of 49 N. This value was adopted because it corresponds to the average value of 15 tests conducted using the equipment shown in Figure 6.

The results from the simulations enable the estimation of deformation, force, and pressure exerted by the FG in the study qualitatively. A Von Mises analysis was not performed because the forces generated by the object are significantly lower than the yield strength of the FG materials.

## 3. Results

A comparative and qualitative analysis between the computational simulations by the FEM and the experimental tests was conducted with the objective of validating the structural characteristics, materials of the FG, and, primarily, the FEM. This analysis was carried out based on the evaluation of gripping forces measured by a load cell and a test specimen.

After validating the FEM, an experimental procedure was carried out to determine the forces applied by the FG during the manipulation of fragile objects. The manipulation of a chicken egg was modeled by the FEM to estimate whether the stresses were suitable for handling an egg, and experiments were then conducted to qualitatively validate the model.

### 3.1. Validation of the FEM Through Experimental Tests

The experimental procedure consisted of measuring the displacement values (closure) of the FG and the respective maximum force applied to a load cell during 15 gripping tests. The load cell used was of the hemispherical type, and a DELIX-brand caliper, model DWKC-2012 (DELIX, Suzhou, China), was used to measure the displacement between the FG fingers. Figure 6 illustrates the FG gripping test on a load cell.

The experiments and computational simulations using the FEM were developed based on the procedures presented in Section 2.2 and Section 2.3, respectively. Table 1 shows the results of these simulations and the experimental tests.

As can be observed in Table 1, the average force and the maximum displacement of the gripper in the experimental tests were 52.66 N and 26.76 mm, respectively, while in the FEM simulation, the displacement value was set to 26.76 mm, which is the average of the experimental values, and the force value was obtained computationally as 49 N. The average percentage difference between the experimental results and the simulation was 7.14%, with the largest difference observed in test 15, at 9.98%, with a standard deviation of 0.81 N for the applied force and 0.32 mm for the displacement. These results indicate that the FEM proposed in Section 2.3 is suitable for representing the dynamic behavior of the gripper, as per the guidelines outlined in [42,43]. These references indicate that, considering the various variables involved in the comparative study between computational simulations and experimental tests, a difference of up to 10% between the values is considered acceptable, validating the FEM as a robust and reliable predictive tool for the subsequent analyses of object manipulation. The overlap of computational and experimental results is presented in Figure 7.

The overlap of the results in Figure 7 demonstrates a strong qualitative and quantitative agreement between the simulated and experimental behavior, further reinforcing the model’s validation.

The FEM algorithms were executed in an average time of 5 min per test. The computer used for the simulations has 64 GB of RAM (Kingston Fury, Kingston Technology, Fountain Valley, CA, USA), a 932 GB SSD (WD Black SN770, Western Digital, San Jose, CA, USA), a 1.8 TB HDD (Western Digital, San Jose, CA, USA), an Intel i7-12700K processor (Intel Corporation, Santa Clara, CA, USA), and a Gigabyte Z690 motherboard (Gigabyte Technology, New Taipei City, Taiwan).

The mesh convergence study, presented in Table 2, was a critical step to ensure that the simulation results are independent of the mesh refinement, thereby guaranteeing the accuracy and reliability of the force predictions. As observed, the relative error decreases considerably as the mesh size is reduced, with variations becoming negligible for values below 1 mm. Therefore, the selection of a 1 mm mesh represents the optimal balance between achieving high computational accuracy and maintaining a reasonable processing time.

### 3.2. Analysis of the Forces Exerted by the FG on a Chicken Egg

The primary advantage of a validated FEM is its ability to analyze complex physical phenomena, such as stress distributions on fragile objects, which are difficult or impossible to measure experimentally without causing damage. To demonstrate this predictive capability, a case study was conducted on the manipulation of a chicken egg. The methodology involved a hybrid approach: first, experimental tests were performed to determine the real-world finger displacement required to grasp the egg. This average displacement was then used as a precise boundary condition in the FEM simulation to quantify the induced stresses and verify the safety of the operation. Visual inspections were also conducted throughout the experiments to confirm that no cracks or fractures occurred.

The experimental procedure involved activating the main motor until the FG fingers grasped the egg, measuring the displacement between the FG fingers for each egg using a caliper, and calculating the average displacement from the three tests. Figure 8 illustrates the manipulation test for one of the eggs used.

As shown in Figure 8, the FG did not cause any type of crack or rupture in the tested egg until the moment the fingers, responsible for securing the egg in the gripper, fully closed. In this experiment, the fingers displaced by 13.3, 14.4, and 15.2 mm, to enable the FG to manipulate the egg, resulting in an average displacement of 14.3 mm across the three tests.

## 4. Discussion

Considering that the proposed FEM is suitable for representing the dynamic behavior of the FG, as discussed in Section 2.3, the force exerted by the FG on the egg was estimated using computational simulations. The mechanical properties of the egg considered in the simulation included a shell elasticity modulus of $E=2.46\times 10^{4}$ MPa and a Poisson’s ratio of μ=0.3, as provided in [31] for an egg with a thickness of 0.34 mm. The other boundary conditions applied in the FEM were mentioned in Section 2.3. Figure 9 presents the results of the FEM simulation.

As illustrated in Figure 9, the maximum tension applied by the gripper on the egg was approximately 7 MPa, while the tension acting on the gripper after the egg manipulation was approximately 14.2 MPa. These results indicate that the material and the design characteristics of the proposed gripper are suitable for handling fragile objects, as the obtained tension values are significantly lower than the material’s yield strength, demonstrating that the gripper does not exhibit brittle failure behavior.

Another important discussion refers to the estimation of the admissible tension for the breaking of an egg. Considering that the average displacement value from the trials with the three eggs was applied to the FEM, it is possible to infer that the egg can resist, on average, a tension of up to 18.3 MPa [31] to be safely manipulated. It is important to emphasize that a preliminary study on the area used for distributing this tension should be conducted.

Although the experimental analyses indicate promising results, the proposed FG presents limitations regarding the manipulation of irregularly shaped objects. These limitations stem from the passive nature of the deformation and the absence of localized control. Moreover, the FG does not employ integrated sensors, such as strain gauges [12,13], which limits its ability to provide real-time force feedback or to automatically adapt to object shapes. These limitations become particularly evident when comparing the proposed FG with more advanced grippers available in the literature. In comparison with other grippers, such as soft pneumatic actuators [12,13] or tendon-driven mechanisms [14,16], the proposed FG stands out due to its simple mechanical design, low manufacturing cost, and ease of fabrication via 3D printing. However, it lacks features such as active stiffness control and contact force regulation, which are commonly found in more complex systems. Given these considerations, it is worth emphasizing that the proposed FG is particularly well-suited for integration into low-cost robotic platforms, especially in applications that demand gentle handling of fragile objects but do not require high precision or closed-loop control.

With respect to the design and analysis methodology, conventional methods for the design and analysis of robotic grippers, which often employ the FEM without direct experimental validation or rely on complex sensors for force control, the framework proposed herein demonstrates an effective integration of simulation and experimental data. This combination enables the estimation of gripping forces with sufficient confidence to ensure the integrity of fragile objects, as demonstrated in the egg case study. Therefore, our method represents a significant advancement by enabling a practical, simplified, and easily implementable analysis, which may contribute to the broader adoption and application of flexible grippers in both industrial and domestic environments.

Furthermore, it is important to note that the present study focused solely on the static behavior of the end-effector during grasping tasks. Dynamic effects resulting from manipulator movement—such as vibrations, inertia, or accelerations—were not taken into account in the current analysis. Future studies should therefore explore the gripper’s performance under dynamic conditions to assess its robustness in real-world applications.

Finally, discrepancies between the simulation and experimental results can be attributed to a variety of factors. These include simplifications in the material model, idealized boundary conditions in the FEM setup, geometric tolerances inherent to the 3D printing process, and minor inaccuracies in the calibration and resolution of the measurement instruments. Despite these sources of error, the deviation remained within an acceptable engineering margin (less than 10%), thereby supporting the validity and consistency of the proposed method.

## 5. Conclusions

The FG was developed with the goal of contributing to studies focused on the development of mechanical grippers for handling fragile objects in domestic, industrial, and other environments. The proposed methods allowed for the validation of the FEM based on data obtained from experimental tests, which enabled the analysis of the FG’s potential for handling fragile objects and estimating the force applied to an egg through a case study.

One of the main objectives of this work was to provide a practical and methodological contribution for other researchers interested in developing flexible grippers. By presenting the stages of development and construction of the FG, the differences compared to other gripper models analyzed in studies [7,11,12,13,14] are highlighted. These differences, including the ability to adapt to objects of different shapes and handle fragile objects, can serve as a foundation for future innovations and improvements in the field. It is important to note that the components of the FG were selected from commercial products, making its construction easier and allowing for new contributions to emerge in the field.

This work also presented a method that establishes a relationship between data obtained experimentally and with the FEM through computational simulation. With the results obtained, other researchers can apply a similar approach to validate different types of grippers and adapt them to various contexts and needs. Experimental procedures were conducted to establish a relationship between the displacement of the FG fingers and the force applied by the gripper to a load cell. After 15 trials conducted to validate the experiment, the results showed a small variation in the standard deviation values of the analyzed sample. The displacement and force results collected experimentally were applied to the ABAQUS software, confirming that the proposed FEM is valid for analyzing the distribution of the gripper’s efforts on the object. Validation was carried out through a comparative analysis between experimental and computational results (Table 1), where the difference between the methods was approximately 7.14%.

The study demonstrates that the proposed Fin Ray-type gripper can safely handle fragile objects, such as chicken eggs, without damaging their shells. The FEM simulation results, consistent with previous studies, show that the gripper applies suitable forces and provides controlled deformation, proving its effectiveness in delicate handling tasks.

Building upon the design and implementation choices described above, this work stands out not only for presenting the physical construction and experimental validation of a simplified Fin Ray gripper, but also for introducing an innovative framework for estimating gripping forces. This framework is supported by an integrated methodology that combines FEM simulations with experimental validation, enabling the safe assessment of fragile object manipulation while reducing reliance on sophisticated sensors and complex computational control. By bridging accessible design with quantitative force estimation, this contribution reinforces the advancement and practical application of flexible grippers in both domestic and industrial contexts.

The authors suggest that future research should aim to validate the application of the FG, or similar mechanical grippers, in handling a wider variety of objects of different sizes, shapes, and fragilities, simulating different industrial and domestic scenarios.

## Figures and Tables

**Figure 1 sensors-25-05782-f001:**
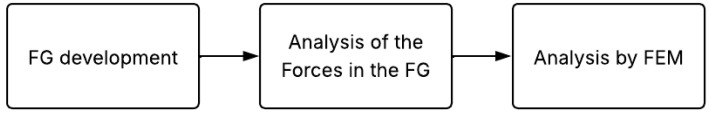
Methods.

**Figure 2 sensors-25-05782-f002:**
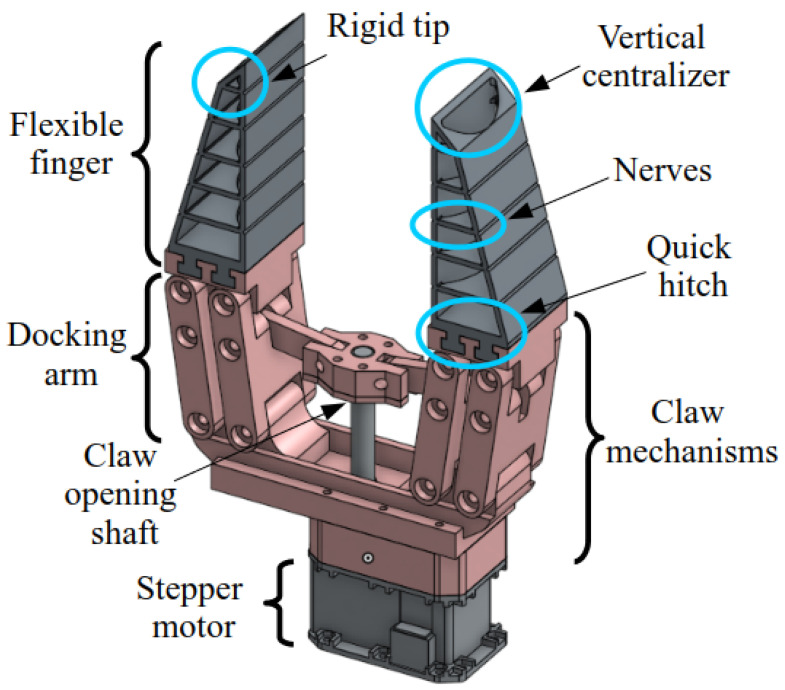
Main components of the FG.

**Figure 3 sensors-25-05782-f003:**
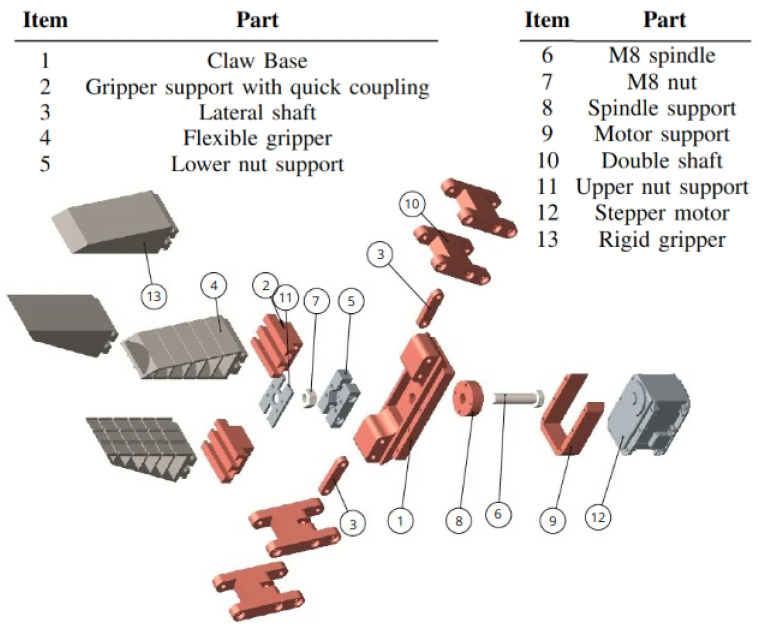
Exploded view of the Fin Ray-type FG.

**Figure 4 sensors-25-05782-f004:**
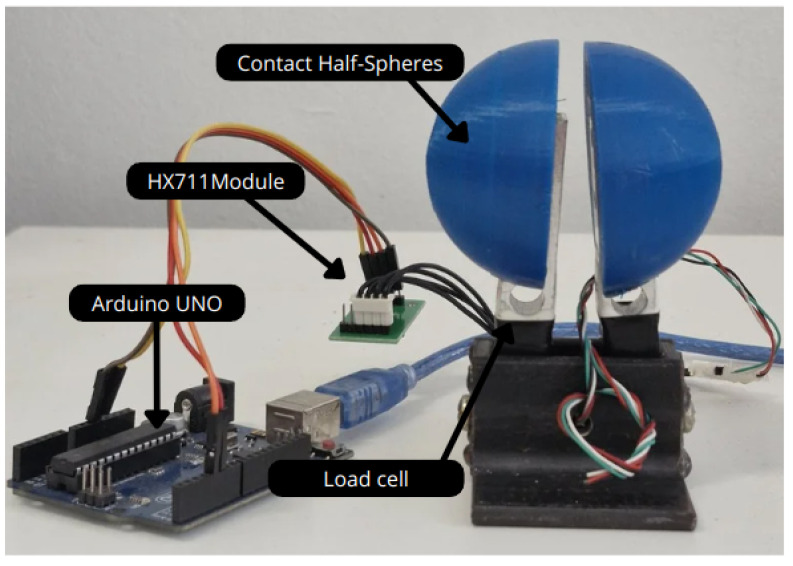
Test bench for the maximum force of the FG.

**Figure 5 sensors-25-05782-f005:**
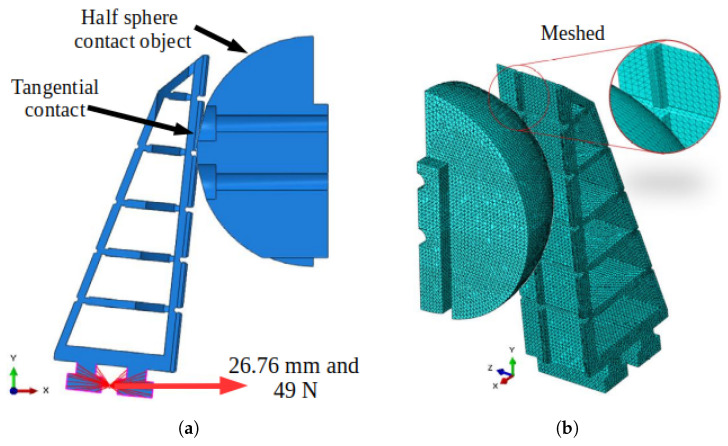
Abaqus environment figures with the restrictions. (**a**) Visualization of restrictions in the Abaqus environment. (**b**) Meshed model representation.

**Figure 6 sensors-25-05782-f006:**
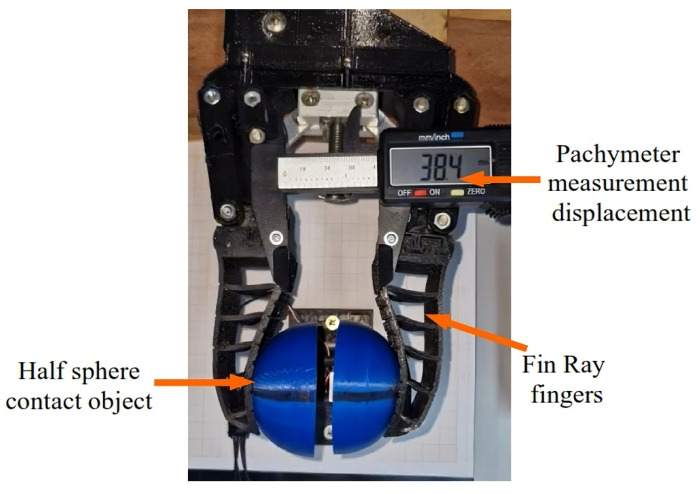
Experimental test on a load cell.

**Figure 7 sensors-25-05782-f007:**
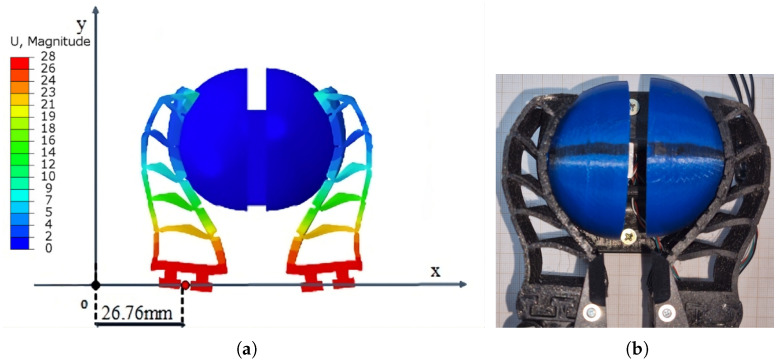
Comparison of experimental and FEM results. (**a**) FEM simulation results. (**b**) Experimental results.

**Figure 8 sensors-25-05782-f008:**
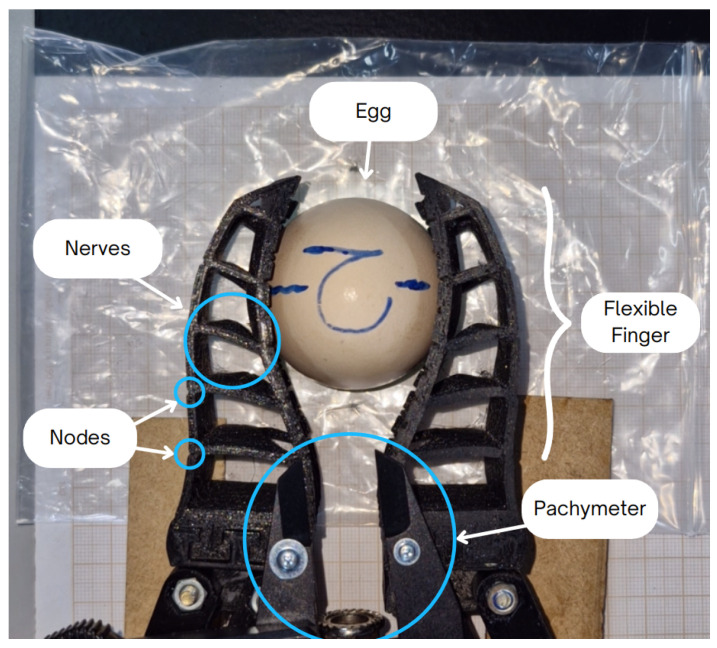
Experiment with a chicken egg.

**Figure 9 sensors-25-05782-f009:**
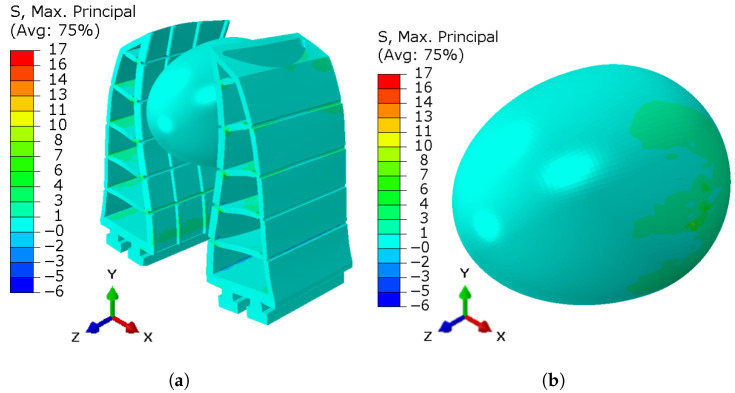
Result of the tension applied by the gripper on the chicken egg. (**a**) Principal stresses on the gripper; (**b**) Principal stresses on the egg.

**Table 1 sensors-25-05782-t001:** Experimental test of r gripping with standard deviation.

Experimental Test			FEM Simulation		
Test	Max. Force [N]	Max. Displacement (mm)	FEM Max. Force [N]	FEM Displacement (mm)	Error (%ε)
1	53.65	26.80	49	26.76	9.57%
2	53.80	26.80	49	26.76	9.80%
3	52.62	26.80	49	26.76	7.35%
4	52.08	26.80	49	26.76	6.31%
5	53.30	26.85	49	26.76	8.63%
6	53.13	26.80	49	26.76	8.43%
7	53.29	26.85	49	26.76	8.73%
8	51.05	26.90	49	26.76	4.18%
9	51.68	25.55	49	26.76	5.47%
10	52.15	26.90	49	26.76	6.49%
11	52.60	26.85	49	26.76	7.35%
12	51.76	26.85	49	26.76	5.63%
13	52.47	26.75	49	26.76	7.14%
14	52.41	26.90	49	26.76	7.06%
15	53.89	26.85	49	26.76	9.98%
Average	52.66	26.76	–	–	7.14%
Standard Deviation	0.8117	0.3235	–	–	–

**Table 2 sensors-25-05782-t002:** Mesh size and relative error.

Mesh Size [mm]	Force [N]	Relative Error [%]
2.00	74.72	–
1.50	57.55	30.00
1.25	53.03	9.00
1.10	50.12	6.00
1.00	49.29	2.00

## Data Availability

The dataset(s) generated during the current study have been deposited in a Google Drive folder (https://drive.google.com/drive/folders/1Z_1t1bxCcxwiSCgG9m2Lw7ysyPtoXsCJ?usp=sharing (accessed on 22 August 2025)) and are publicly available.

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
