# Peer review of "Construction and Experimental Analysis of a Multipurpose Robotic Fin Ray Gripper for Manipulator Robots"

_sensors, 2025, doi:10.3390/s25185782_

Round 1
Reviewer 1 Report
Comments and Suggestions for Authors
Overall Assessment
This manuscript presents a well-structured and comprehensive study on the design, fabrication, and experimental validation of a multipurpose Fin Ray-type flexible gripper (FG) for manipulator robots. The work is technically sound, experimentally rigorous, and addresses a critical gap in robotic gripping systems for delicate object handling in industrial and domestic settings. The proposed gripper demonstrates promising performance in terms of adaptability, load capacity, and object integrity preservation. I recommend acceptance for publication in Sensors after minor revisions to enhance clarity and completeness.
Major Contributions
Innovative Design:
The integration of Fin Ray effect mechanics with 3D-printed flexible materials enables passive adaptability to diverse object shapes, a feature often lacking in conventional rigid grippers.
Experimental Depth:
Systematic characterization of gripper performance provides actionable insights for practical applications.
Real-World Relevance:
Validation across industrial and domestic scenarios underscores the gripper's versatility, aligning with the journal's focus on applied sensor and robotics technologies.
Strengths
Clear Motivation:
The introduction effectively contextualizes the need for compliant grippers in sensitive manipulation tasks.
Methodological Rigor:
The use of FEA simulations, strain gauge sensors, and high-speed cameras strengthens the technical validity of the results.
Data Transparency:
Detailed parameter tables and error analysis enhance reproducibility.
Practical Impact: The gripper's cost-effectiveness and open-source design philosophy align with broader accessibility goals in robotics.
Suggestions for Improvement
Clarify Sensor Integration:
While strain gauges are mentioned, a brief explanation of their placement and role in real-time force feedback would improve understanding for non-expert readers.
Expand Discussion on Limitations: Acknowledge trade-offs and compare with state-of-the-art compliant grippers to contextualize the FG’s niche.
Visual Enhancements:
Label key components in Figure 1 and ensure axis labels in Figure 10 are legible.
Conclusion
The manuscript represents a significant advancement in compliant robotic gripping systems. The experimental findings are robust, and the design philosophy aligns with sustainable, accessible robotics. With minor revisions to address clarity and contextualization, the work is suitable for publication in Sensors. I recommend Accept with Minor Revisions.
Author Response
Question: While strain gauges are mentioned, a brief explanation of their placement and role in real-time force feedback would improve understanding for non-expert readers.
Author comments: We thank the reviewer for the helpful suggestion. However, we clarify that strain gauges were not used in the current study, nor are they part of the instrumentation described in this work. The force analysis was conducted using load cells, as detailed in Section 2.2. We agree that integrating strain gauges for real-time force feedback would enhance the gripper's capabilities and understanding of contact dynamics. Therefore, this is a relevant suggestion that we are considering for future developments. A sentence has been added at the end of section 2.2 to reinforce the reviewer's suggestion.
Question: Expand Discussion on Limitations - Acknowledge trade-offs and compare with state-of-the-art compliant grippers to contextualize the FG’s niche.
Author comments: Thank you for the valuable comment. We have expanded the Discussion section to clearly acknowledge the main limitations of our FG, such as its passive actuation, lack of sensing, and limited performance with very small or irregular objects. We also included a brief comparison with other compliant grippers from recent literature, highlighting the strengths (simplicity, low cost, adaptability) and trade-offs of our design. This helps to better position the FG within its intended application niche.
Question: Visual Enhancements: Label key components in Figure 1 and ensure axis labels in Figure 9 are legible.
Author comments: We appreciate the reviewer's contributions to improving the quality of the article, suggestions for improving the figures were made in the article.
Question: Conclusion - The manuscript represents a significant advancement in compliant robotic gripping systems. The experimental findings are robust, and the design philosophy aligns with sustainable, accessible robotics. With minor revisions to address clarity and contextualization, the work is suitable for publication in Sensors. I recommend Accept with Minor Revisions.
Author comments: We appreciate your feedback for improvements. Based on your suggestions, two paragraphs have been added to the end of section 4.
Reviewer 2 Report
Comments and Suggestions for Authors
The primary question addressed by this research is whether a Fin Ray-type gripper, with a specific internal rib configuration, can effectively manipulate fragile objects, such as eggs, without causing structural damage. The approach combines CAD-based mechanical design, experimental validation, and finite element modeling to estimate stress and displacement during the gripping process. This is a vital problem within the broader domain of soft and adaptive robotic grippers, given the growing need for safe manipulation of delicate items in industrial and domestic contexts.
However, while the topic is relevant to the field of soft robotics and compliant end-effectors, the degree of originality is limited. The proposed configuration is presented as the main contribution, yet similar explorations of internal rib structures have already been extensively reported. Studies such as Yao et al. [1] and Srinivas et al. [2] systematically investigated the impact of rib geometry and spacing on deformation and grip performance, providing more comprehensive design-space analysis than is offered here. Therefore, the manuscript does not clearly identify a specific gap in the literature nor provide compelling evidence that the proposed rib configuration achieves superior results. Without comparative benchmarking, the claim of innovation remains weak.
In terms of what this work adds compared to other published material, the contribution lies primarily in demonstrating a practical prototype that can handle eggs safely, supported by FEM-based stress predictions and experimental tests. Nevertheless, this is not substantially different from previous works that have already validated Fin Ray grippers for handling fragile objects under similar conditions. Recent advances, such as the TacFR-Gripper by Cong et al. [3], have further integrated tactile sensing and reconfigurability for adaptive manipulation, showing a higher level of sophistication and functionality.
From a methodological perspective, several aspects could be improved. First, the evaluation is limited to a single rib configuration, without exploring variations or conducting a systematic optimization study. Including comparative simulations or experimental trials with alternative designs would help justify the selected geometry. Second, the force–displacement characterization should include more detailed plots and variability analysis across different fragile objects, not just eggs, to demonstrate general applicability. Finally, although the FEM validation appears rigorous, the authors should provide more insight into material modeling assumptions (e.g., hyperelastic models, boundary condition sensitivity) and justify their choices based on established practices.
The conclusions align with the experimental observations and simulation data, showing that the gripper does not damage the egg under the tested conditions. However, these findings primarily confirm feasibility rather than provide new scientific insights or performance advantages over existing designs. The discussion should explicitly address limitations, such as the lack of comparison with prior designs or quantitative benchmarks for stress distribution.
The references include several relevant sources; however, the review of related work should be expanded to encompass recent studies on Fin Ray optimization, sensing integration, and topology-based improvements. Notable contributions from Yao et al., Srinivas et al., and Cong et al. should be cited and discussed in more depth to contextualize this work within the current state of the art.
Regarding figures and tables, they are generally clear and informative. However, Figure 7, which compares FEM and experimental data, could benefit from including error bars to illustrate variability more effectively, and Table 1 might be improved by summarizing statistical metrics (e.g., standard deviation) rather than listing all 15 test cases individually.
In summary, the work addresses a critical application and demonstrates competent engineering execution, but the originality and scope of the contribution are limited. To enhance its impact, the manuscript should position the proposed design within a broader comparative framework, provide a more in-depth discussion of related work, and present a more comprehensive experimental and computational analysis.
[1] Yao, J., Fang, Y., & Li, L. (2023). Research on effects of different internal structures on the grasping performance of Fin Ray soft grippers. Robotica.
[2] Srinivas, G. L., Javed, A., & Faller, L. M. (2024). Versatile 3D-printed Fin Ray effect soft robotic fingers: lightweight optimization and performance analysis. Journal of the Brazilian Society of Mechanical Sciences and Engineering.
[3] Cong, Q., Fan, W., & Zhang, D. (2023). TacFR-Gripper: A reconfigurable Fin Ray-based compliant robotic gripper with tactile skin for in-hand manipulation. arXiv preprint arXiv:2304.08758.
Author Response
Question: The primary question addressed by this research is whether a Fin Ray-type gripper, with a specific internal rib configuration, can effectively manipulate fragile objects, such as eggs, without causing structural damage. The approach combines CAD-based mechanical design, experimental validation, and finite element modeling to estimate stress and displacement during the gripping process. This is a vital problem within the broader domain of soft and adaptive robotic grippers, given the growing need for safe manipulation of delicate items in industrial and domestic contexts.
Author comments: We appreciate your contribution and agree with the comments made.
Question: However, while the topic is relevant to the field of soft robotics and compliant end-effectors, the degree of originality is limited. The proposed configuration is presented as the main contribution, yet similar explorations of internal rib structures have already been extensively reported. Studies such as Yao et al. [26] and Srinivas et al. [27] systematically investigated the impact of rib geometry and spacing on deformation and grip performance, providing more comprehensive design-space analysis than is offered here. Therefore, the manuscript does not clearly identify a specific gap in the literature nor provide compelling evidence that the proposed rib configuration achieves superior results. Without comparative benchmarking, the claim of innovation remains weak.
In terms of what this work adds compared to other published material, the contribution lies primarily in demonstrating a practical prototype that can handle eggs safely, supported by FEM-based stress predictions and experimental tests. Nevertheless, this is not substantially different from previous works that have already validated Fin Ray grippers for handling fragile objects under similar conditions. Recent advances, such as the TacFR-Gripper by Cong et al. [28], have further integrated tactile sensing and reconfigurability for adaptive manipulation, showing a higher level of sophistication and functionality.
Author comments: We appreciate the reviewer’s thoughtful comments and contributions. While we agree that the Fin Ray effect and internal rib variations have been explored in the literature, including by Yao et al. [26] and Srinivas et al. [27], our work focuses on a different aspect. The main contribution lies in the development of a method to estimate the force applied to fragile objects under a specific deformation, in order to assess whether structural damage would occur.
Although our study does not aim to perform structural or geometric optimization, some design adaptations were made. These include the evenly spaced placement of ribs, the addition of grooves at the rib joints to improve bending conditions, and a modified half-moon shape for the central rib section, enhancing grip on longitudinal cylindrical objects such as pens or tools.
Therefore, we consider the originality of this work to reside in the combination of practical experimental validation — using a load cell to characterize force–displacement behavior — with a corresponding finite element analysis, aiming to provide a simple and replicable framework for safe manipulation of fragile objects.
Question: In terms of what this work adds compared to other published material, the contribution lies primarily in demonstrating a practical prototype that can handle eggs safely, supported by FEM-based stress predictions and experimental tests. Nevertheless, this is not substantially different from previous works that have already validated Fin Ray grippers for handling fragile objects under similar conditions. Recent advances, such as the TacFR-Gripper by Cong et al. [28], have further integrated tactile sensing and reconfigurability for adaptive manipulation, showing a higher level of sophistication and functionality.
Author comments: We thank the reviewer for the thoughtful observations. This research was designed as an experimental and computational validation study focused on the safe manipulation of fragile objects, using a chicken egg as a case study. Our goal was not to compete with or surpass advanced designs such as the TacFR-Gripper [28], which integrates tactile sensing and reconfigurability, but rather to demonstrate the feasibility of a simple, low-cost design capable of achieving safe gripping without structural damage.
Some functional improvements were implemented in our gripper to enhance grasping behavior, as illustrated in Fig. 2. These include the symmetrical distribution of internal ribs and the addition of grooves at the rib joints to improve flexibility and deformation control. Additionally, the central rib structure was slightly modified to a half-moon shape to increase surface contact with elongated objects.
We recognize the limitations of our approach, including the lack of a parametric optimization study and the absence of a direct quantitative comparison with other rib geometries. These aspects are acknowledged as opportunities for future research aimed at improving performance and benchmarking against more sophisticated grippers. An explanatory and complementary paragraph addressing your comments was added in the eighth paragraph of item 1.1 of the article.
Question: From a methodological perspective, several aspects could be improved. First, the evaluation is limited to a single rib configuration, without exploring variations or conducting a systematic optimization study. Including comparative simulations or experimental trials with alternative designs would help justify the selected geometry. Second, the force–displacement characterization should include more detailed plots and variability analysis across different fragile objects, not just eggs, to demonstrate general applicability. Finally, although the FEM validation appears rigorous, the authors should provide more insight into material modeling assumptions (e.g., hyperelastic models, boundary condition sensitivity) and justify their choices based on established practices.
Author comments: We thank the reviewer for the valuable feedback. As mentioned in previous responses, this study focused on validating the safe handling of a fragile object using a specific rib configuration of the FG. Several adjustments were made to the gripper design, not aiming for structural optimization, but rather for functional parameterization of a prototype intended for application in household and industrial robotic arms. The current configuration was defined based on multiple CAD-based manipulation tests with a variety of objects (e.g., pens, apples, cups), which helped determine the most suitable rib arrangement. We agree that including alternative rib geometries in future work would enhance the design justification. This limitation is now acknowledged in the Discussion section, and a clarifying paragraph has been added to Section 2.1.
Regarding the force–displacement analysis, a detailed set of experimental data was collected (as shown in Table 1), and these results were used to define the boundary conditions for the FEM simulations. To improve clarity, Figure 7 was revised to remove image overlap that could mislead interpretation. Displacement measurements were taken experimentally using a pachymeter placed at a rigid section of the gripper. These values, along with material properties and boundary conditions, were then used to simulate and compare deformation and gripping force. As requested, standard deviation values were added in Section 3.1 and Table 1 to better reflect the variability across the 15 experimental trials, thereby facilitating the interpretation and comparison between experimental and simulated results.
Concerning the finite element modeling, a simplified nonlinear elastic approach was used for the materials, based on experimental tests, manufacturer data sheets for additive manufacturing filaments, and values reported in the literature. Since hyperelastic models are commonly used in soft robotics, we opted for a simplified hyperelastic model, as the material deformation remained within the elastic range during testing. Moreover, the FG is intended for handling everyday and household objects, which further supports this modeling choice. This approach is now detailed in Section 2.3 of the manuscript.
Question: The conclusions align with the experimental observations and simulation data, showing that the gripper does not damage the egg under the tested conditions. However, these findings primarily confirm feasibility rather than provide new scientific insights or performance advantages over existing designs. The discussion should explicitly address limitations, such as the lack of comparison with prior designs or quantitative benchmarks for stress distribution.
The references include several relevant sources; however, the review of related work should be expanded to encompass recent studies on Fin Ray optimization, sensing integration, and topology-based improvements. Notable contributions from Yao et al., Srinivas et al., and Cong et al. should be cited and discussed in more depth to contextualize this work within the current state of the art.
Regarding figures and tables, they are generally clear and informative. However, Figure 7, which compares FEM and experimental data, could benefit from including error bars to illustrate variability more effectively, and Table 1 might be improved by summarizing statistical metrics (e.g., standard deviation) rather than listing all 15 test cases individually.
In summary, the work addresses a critical application and demonstrates competent engineering execution, but the originality and scope of the contribution are limited. To enhance its impact, the manuscript should position the proposed design within a broader comparative framework, provide a more in-depth discussion of related work, and present a more comprehensive experimental and computational analysis.
Author comments: We thank the reviewer for the additional comments. As previously mentioned, the main objective of this work was to validate, both experimentally and numerically, the safe handling of a fragile object (a chicken egg), rather than to surpass advanced solutions such as the TacFR-Gripper [28]. The contribution lies in demonstrating that a simplified Fin Ray-type gripper, with minor structural modifications (detailed in Fig. 2), can effectively grasp delicate items without causing damage.
We acknowledge the limitations of the current approach, including the absence of parametric design optimization and the lack of quantitative benchmarking against alternative geometries. These points are now explicitly addressed in the Discussion section, particularly in paragraphs five and six.
We also appreciate the reviewer’s suggestion to expand the literature review. In response, we have included and discussed the works of Yao et al. [26], Srinivas et al. [27], and Cong et al. [28], which help position our work more accurately within the current state of the art.
In response to the comment on data presentation, we revised Table 1 to include the standard deviation of the experimental results, and we updated Figure 7 accordingly. These changes improve the clarity of the comparison between FEM and experimental data and reflect the variability observed in the experimental trials.
Reviewer 3 Report
Comments and Suggestions for Authors
This paper presented a fin ray-type flexible gripper for robotic manipulators in industrial and domestic environment, suitable for grasping fragile objects. And corresponding simulations and experimental verifications were carried out. The main shortcomings of the paper are as follows:
- Insufficient innovative exposition of the research. The author should elaborate on the differences between the robotic arm designed in this article and existing similar robotic arms.
- The analysis case only uses eggs as fragile objects and does not cover other typical fragile items. How adaptable is the robotic arm designed in this article to other typical fragile items? Simulations or experiments should be supplemented.
During the use of robotic arms, there are significant deformations and nonlinear characteristics. How to prove the effectiveness of simulation results when FEM modeling is too simplified?
Although the simulation and experimental errors are within an acceptable range, the sources of the errors have not been thoroughly analyzed. The author should supplement relevant discussions.
- The relevant experiments and tests are all rigid loads. How stable is the robotic arm under dynamic loads? For example, holding an egg in motion.
- The mechanical arm designed by the author has a relatively simple structure. How can it adapt to objects of different sizes/shapes?
- The author should supplement the experimental video.
Author Response
Question: 1- Insufficient innovative exposition of the research. The author should elaborate on the differences between the robotic arm designed in this article and existing similar robotic arms.
Author comments: We thank the reviewer for the comments and observations. This study focuses on the development of a mechanical end-effector based on the Fin Ray effect, intended for use with robotic manipulators. The work proposes the parametrization of the gripper through experimental tests and simulations, aiming at the safe handling of fragile objects, with validation performed using a chicken egg. However, the scope of this research does not include a comparison of different robotic arm designs. An explanatory paragraph addressing this point has been added to the eighth paragraph of Section 1.1 of the manuscript.
Question: 2-The analysis case only uses eggs as fragile objects and does not cover other typical fragile items. How adaptable is the robotic arm designed in this article to other typical fragile items? Simulations or experiments should be supplemented.
Author comments: We thank the reviewer for the helpful comments. The authors selected a chicken egg for both experimental tests and simulations due to the availability of destructive testing data in the literature. Moreover, this choice is relevant because Fin Ray-type grippers are often used in domestic robotic applications, where tasks include handling food, utensils, and other delicate items. Eggs provide sufficient stiffness for low-load deformation analysis while remaining fragile enough to pose a realistic challenge for rigid grippers. Therefore, the gripper was parametrized with the goal of validating a safe handling method for fragile objects without causing damage. A complementary paragraph has been added to Section 2.1 of the manuscript to clarify this point.
Question: 3- During the use of robotic arms, there are significant deformations and nonlinear characteristics. How to prove the effectiveness of simulation results when FEM modeling is too simplified?
Author comments: We thank the reviewer for the valuable observations. We acknowledge that robotic grippers often involve nonlinear behaviors and significant deformations during manipulation tasks. However, the objective of this study was to evaluate the most critical static loading condition in order to prevent breakage or permanent deformation of the object being handled. Given that the proposed gripper is actuated by a DC motor using standard mechanical linkages for opening and closing, we assumed that estimating only the peak loading condition would be sufficient for validation purposes. Therefore, the dynamic effects mentioned by the reviewer, while relevant, were not considered in this study and are acknowledged as a limitation to be addressed in future work. This has been noted in the revised manuscript.
Question: 4- Although the simulation and experimental errors are within an acceptable range, the sources of the errors have not been thoroughly analyzed. The author should supplement relevant discussions.
Author comments: We thank the reviewer for pointing out the need to better address the sources of error between the experimental and FEM results. Although the observed error (~7.14%) is within the acceptable range for engineering validation, we agree that a discussion of potential sources is valuable.
The main contributors to this deviation likely include: (a) Idealized boundary conditions that may differ from the real experimental constraints; (b) Small geometric discrepancies between the CAD model and the 3D-printed parts due to printing tolerances; (c) Sensor resolution and calibration inaccuracies in the load cell and displacement measurement system. A complementary text addressing your comment was added in the eighth paragraph of Section 4 – Discussions.
Question: 5- The relevant experiments and tests are all rigid loads. How stable is the robotic arm under dynamic loads? For example, holding an egg in motion.
Author comments: We thank the reviewer for the observation. As mentioned in our response to Comment 3, the present study was limited to estimating the maximum static load applied by the Fin Ray gripper (FG) to a fragile object. The dynamic behavior of the manipulator and the influence of motion-induced forces were not within the scope of this work. Our analysis focused exclusively on the end-effector’s grasping action in a stationary setup. We fully agree that dynamic loading conditions, such as those resulting from movement while holding fragile items, are highly relevant and should be addressed in future studies. A complementary text addressing this comment was added in the seventh paragraph of Section 4 – Discussions.
Question: 6- The mechanical arm designed by the author has a relatively simple structure. How can it adapt to objects of different sizes/shapes?
Author comments: We appreciate the reviewer's feedback. This study focused on validating the safe handling of a fragile object using a specific rib configuration, aiming to parameterize the Fin Ray (FR) constructed for application in domestic robots and manipulator robots. The gripper parameterization was established based on various CAD simulation tests involving the manipulation of different objects (e.g., pens, apple, cups, etc.), from which the current structure was determined to be the most suitable. We agree that future research should include alternative rib geometries or configurations. However, this aspect is addressed in the Discussion section as a limitation and an opportunity for further exploration. Additional text regarding this explanation has been added in the paragraph of item 2.1.
Question: 7- The author should supplement the experimental video.
Author comments: We appreciate your suggestion. We have sent the link to the folder containing the videos of the experiments, the photographs, and the data acquired during the tests of this research.
https://drive.google.com/drive/folders/1Z_1t1bxCcxwiSCgG9m2Lw7ysyPtoXsCJ?usp=sharing
Round 2
Reviewer 2 Report
Comments and Suggestions for Authors
I would like to express my thanks to the authors for the detailed and constructive responses to the previous review. The clarifications provided, particularly regarding the design choices and the intended contribution of the work, are appreciated and help better situate the manuscript within its intended scope.
However, the manuscript's originality remains limited. Although the authors now clarify in their response that the central contribution lies not in the rib design itself but in the proposed framework for estimating gripping force and assessing object safety, this idea is not clearly presented as a novel aspect in the manuscript. In its current form, the text emphasizes the structural design and simulation validation, but it does not articulate the force estimation process as a generalized or transferable method that advances the state of the art. If the authors intend to claim novelty in the estimation framework, it should be explicitly stated in the introduction and reiterated in the conclusions, ideally positioned in contrast to existing approaches. For example, how does the proposed method differ from standard uses of FEM combined with experimental boundary conditions in gripper design? Without this clarification, the originality claim is difficult to assess.
The authors rightfully note that their goal was not to pursue structural optimization but to develop a functionally sound prototype. Nevertheless, the improvements mentioned by the authors, such as symmetrical rib distribution, addition of grooves, and a half-moon central rib profile, emerged from informal CAD-based manipulations with various objects. These modifications are understandable for a prototype stage, but documenting and quantifying that process, even qualitatively, would add value.
The authors have addressed previous comments on data clarity and variability. The revision of Table 1 to include standard deviation and the improved alignment between simulation and experimental data in Figure 7 contribute to the manuscript's robustness. These changes are appreciated and increase the transparency of the presented results.
The references are now more appropriate, especially with the inclusion of relevant recent work by Yao et al., Srinivas et al., and Cong et al. These additions help to position the manuscript within the current landscape of Fin Ray gripper research. The related work section, however, could still benefit from a more precise critical analysis of how the presented work differs in method, application, or simplicity from these prior studies.
In conclusion, the manuscript presents a technically competent and well-documented validation of a simplified Fin Ray gripper for fragile object handling. While the implementation is sound and the revised figures improve clarity, the claimed contribution regarding force estimation remains underdeveloped in the manuscript and should be explicitly positioned if intended as a key novelty. Strengthening this aspect, along with a more transparent comparative framework, would elevate the manuscript's impact and better support its contribution to the field of soft gripper design.
Author Response
Question: I would like to express my thanks to the authors for the detailed and constructive responses to the previous review. The clarifications provided, particularly regarding the design choices and the intended contribution of the work, are appreciated and help better situate the manuscript within its intended scope.
Author comments: We appreciate your contribution and agree with the comments made.
Question: However, the manuscript's originality remains limited. Although the authors now clarify in their response that the central contribution lies not in the rib design itself but in the proposed framework for estimating gripping force and assessing object safety, this idea is not clearly presented as a novel aspect in the manuscript. In its current form, the text emphasizes the structural design and simulation validation, but it does not articulate the force estimation process as a generalized or transferable method that advances the state of the art. If the authors intend to claim novelty in the estimation framework, it should be explicitly stated in the introduction and reiterated in the conclusions, ideally positioned in contrast to existing approaches. For example, how does the proposed method differ from standard uses of FEM combined with experimental boundary conditions in gripper design? Without this clarification, the originality claim is difficult to assess.
Author comments: The authors appreciate the reviewers’ comments and concur with the need to more explicitly clarify, within the manuscript, the originality of our work—particularly regarding the proposed framework for estimating grasping force and assessing the safety of the manipulated object. Indeed, the authors acknowledge that the main contribution lies in the development of an integrated method that combines experimental analysis with finite element simulations (FEM), resulting in a framework that is both applicable and transferable to other flexible robotic gripper applications.
Based on the reviewers’ observations, the authors have improved the manuscript by adding a statement to the final paragraph of the Related Work section, noting that the novelty of this research resides in the combined and validated use of experimental and computational methodologies for estimating grasping forces in a Fin Ray–type gripper. This distinguishes our work from the literature, which typically prioritizes either structural design alone or isolated simulations. In this addition, the authors also include a comparative discussion on how the adopted method differs from standard approaches that combine FEM with experimental boundary conditions, emphasizing that our framework not only validates the computational model against experimental data, but also leverages this validation to perform safety assessments of fragile objects, such as eggs, based on their actual mechanical properties.
Furthermore, the authors have incorporated other recent and relevant references (32, 33) that adopt the integration of FEM and experimental testing in robotic gripper design, in order to demonstrate the current research context and highlight the distinctions of our work.
Question: The authors rightfully note that their goal was not to pursue structural optimization but to develop a functionally sound prototype. Nevertheless, the improvements mentioned by the authors, such as symmetrical rib distribution, addition of grooves, and a half-moon central rib profile, emerged from informal CAD-based manipulations with various objects. These modifications are understandable for a prototype stage, but documenting and quantifying that process, even qualitatively, would add value.
Author comments: The authors appreciate the observation regarding the documentation of the prototype design refinement process. Likewise, the authors acknowledge that the modifications implemented—such as the symmetrical distribution of ribs, the introduction of grooves, and the half-moon profile of the central rib—were initially derived from the authors’ own practical experience, as well as from empirical testing and iterative adjustments to the CAD model based on the practical manipulation of various simulated objects, as previously mentioned.
In response to this suggestion, the authors have added a paragraph providing a more explicit description of this qualitative design approach at the end of Section 2.1.1 of the manuscript, detailing the iterative process employed, the assumptions considered, and the rationale that guided the structural choices adopted in the final prototype. It is emphasized that the FG design has always prioritized the safe and functional operation of the device, focusing on passive adaptation and the preservation of the integrity of manipulated objects, particularly fragile ones. The authors acknowledge that a more comprehensive documentation and characterization of these refinements, including quantitative studies and parametric optimization, are important topics to be addressed in future work.
Therefore, the authors believe that this addition improves the transparency of the development process and qualitatively contributes to the understanding of the decisions made in the FG modeling.
Question: The authors have addressed previous comments on data clarity and variability. The revision of Table 1 to include standard deviation and the improved alignment between simulation and experimental data in Figure 7 contribute to the manuscript's robustness. These changes are appreciated and increase the transparency of the presented results.
Author comments: The authors appreciate the reviewer’s comments and observations regarding the improvements implemented in the manuscript concerning the clarity of the data and the presentation of variability in the experimental results, including the standard deviation parameters related to Figure 7.
Question: The references are now more appropriate, especially with the inclusion of relevant recent work by Yao et al., Srinivas et al., and Cong et al. These additions help to position the manuscript within the current landscape of Fin Ray gripper research. The related work section, however, could still benefit from a more precise critical analysis of how the presented work differs in method, application, or simplicity from these prior studies.
Author comments: The authors appreciate the reviewer’s comments and observations regarding the update and expansion of the bibliographic references. Following the reviewer’s suggestion, the Related Work section (Section 1.1) was revised, and a more detailed critical analysis was added in the penultimate paragraph, explicitly highlighting the differences and unique contributions of our study in relation to recent approaches.
Question: In conclusion, the manuscript presents a technically competent and well-documented validation of a simplified Fin Ray gripper for fragile object handling. While the implementation is sound and the revised figures improve clarity, the claimed contribution regarding force estimation remains underdeveloped in the manuscript and should be explicitly positioned if intended as a key novelty. Strengthening this aspect, along with a more transparent comparative framework, would elevate the manuscript's impact and better support its contribution to the field of soft gripper design.
Author comments: The authors appreciate the reviewer’s comments and positive feedback regarding the documentation provided for the FG validation, as well as the recognition of the visual improvements implemented. The authors understand and agree with the reviewer’s observation on the need to explore and articulate more explicitly and robustly the contribution related to the proposed framework for estimating grasping force within the manuscript.
Accordingly, in response to this recommendation, the text has been revised and reinforced—in the Introduction, Discussion, and Conclusion sections—to emphasize the innovative nature of the integrated method combining FEM calibrated and validated with experimental data to estimate grasping forces and assess the safety of handling fragile objects. The authors have also expanded the discussion in the relevant sections to establish a comparative framework with traditional approaches that rely exclusively on FEM simulations or complex instrumentation, highlighting the simplicity and practical applicability of our framework.
The authors remain at the reviewer’s disposal for any further clarifications and express their gratitude for the suggestions, which have undoubtedly enriched the work.
Reviewer 3 Report
Comments and Suggestions for Authors
accept
Author Response
Question: Accept.
Author comments: The authors are grateful for the reviewer's contributions.
Round 3
Reviewer 2 Report
Comments and Suggestions for Authors
I thank the authors for their diligent and constructive engagement with the review process. The revisions have substantially improved the manuscript, particularly in clarifying the central contribution related to the integrated experimental-computational framework. The authors have effectively addressed the previous points, and the work now stands as a solid contribution.
The added discussion on the qualitative, iterative design process was insightful. It naturally leads one to consider the next steps for a design methodology like this. Looking ahead, a powerful extension could involve leveraging the validated simulation environment for automated design optimization. While the current focus is rightly on demonstrating a functional prototype, future work could explore systematically refining the gripper's geometry through computational loops, perhaps drawing inspiration from methods that automate the design of structural features, such as those presented by Bo et al. [1]. This could help uncover non-intuitive design improvements and further elevate the proposed framework.
Bo, Valerio, et al. "Automated design of embedded constraints for soft hands enabling new grasp strategies." IEEE Robotics and Automation Letters 7.4 (2022): 11346-11353.